# Investigation of the General Molecular Mechanisms of Gallic Acid via Analyses of Its Transcriptome Profile

**DOI:** 10.3390/ijms25042303

**Published:** 2024-02-15

**Authors:** Jiyeon Kim, Bo Kyung Kim, Sang Hyun Moh, Goo Jang, Jae Yong Ryu

**Affiliations:** 1Laboratory of Theriogenology and Biotechnology, Department of Veterinary Clinical Science, College of Veterinary Medicine and the Research Institute of Veterinary Science, Seoul National University, Seoul 08826, Republic of Korea; jykim.snu@snu.ac.kr; 2Plant Cell Research Institute of BIO-FD&C Co., Ltd., Incheon 21990, Republic of Korea; biofdnc@gmail.com; 3Department of Biotechnology, Duksung Women’s University, 33 Samyang-Ro 144-Gil, Dobong-gu, Seoul 01369, Republic of Korea; bokyung0617@duksung.ac.kr

**Keywords:** gallic acid, transcriptome, transcriptome profile changes

## Abstract

Gallic acid (GA), a phenolic compound naturally found in many plants, exhibits potential preventive and therapeutic roles. However, the underlying molecular mechanisms of its diverse biological activities remain unclear. Here, we investigated possible mechanisms of GA function through a transcriptome-based analysis using LINCS L1000, a publicly available data resource. We compared the changes in the gene expression profiles induced by GA with those induced by FDA-approved drugs in three cancer cell lines (A549, PC3, and MCF7). The top 10 drugs exhibiting high similarity with GA in their expression patterns were identified by calculating the connectivity score in the three cell lines. We specified the known target proteins of these drugs, which could be potential targets of GA, and identified 19 potential targets. Next, we retrieved evidence in the literature that GA likely binds directly to DNA polymerase β and ribonucleoside-diphosphate reductase. Although our results align with previous studies suggesting a direct and/or indirect connection between GA and the target proteins, further experimental investigations are required to fully understand the exact molecular mechanisms of GA. Our study provides insights into the therapeutic mechanisms of GA, introducing a new approach to characterizing therapeutic natural compounds using transcriptome-based analyses.

## 1. Introduction

Gallic acid (GA) is a polyphenol compound naturally present in various plant species, such as gallnut, sumac, tea leaves, and oak bark [1]. This phytochemical demonstrates different biological features, including antioxidant, anti-inflammatory [2], antidepressant [3], and antidiabetic properties [4]. Furthermore, GA may play therapeutic roles in various diseases caused by oxidative stress, such as cancer [5], cardiovascular diseases [6], and neurodegenerative disorders [7]. Additionally, GA modulates cell signaling pathways in different cancers, including leukemia and liver, lung, colon, prostate, and breast cancers. In particular, GA regulates cell proliferation and survival, induces apoptosis, inhibits angiogenesis, and triggers oxidative stress [8,9,10,11,12]. However, the underlying molecular mechanism of GA remains unclear, since GA may exert its effects via multiple mechanisms.

Previous studies have used omics data to explain the action mechanisms of chemical compounds. For instance, transcriptome analyses have been widely used to facilitate the identification of differentially expressed genes (DEGs) based on their expression and regulatory mechanisms [13,14]. Recently, publicly available transcriptional profiling databases such as Connectivity Map (CMap) have emerged, dramatically improving the interpretation of transcriptional analysis [15]. CMap provides information on the gene expression profiles (GEPs) obtained from human cancer cells subjected to drug treatment, facilitating the identification of drug–disease associations using pattern-matching algorithms. Since 2006, CMap has compiled a library containing the GEPs of 564 genes for 143 different small-molecule perturbagens. The Library of Integrated Network-based Cellular Signatures (LINCS) L1000 database, an improved version of CMap, features considerably more information because it covers 1127 distinct cell lines subjected to drug treatment, encompassing approximately 42,000 small molecules (https://lincsportal.ccs.miami.edu/dcic-portal/, accessed on 12 January 2023) [16]. This database serves as a valuable resource for comparing GEP changes triggered by drug treatment in different disease models [17], predicting drug-target interactions [18,19], network integration [20], and transcriptomic analysis to clarify the mechanism of actions of bioactive molecules [21,22].

In this study, we analyzed the potential mechanism of action of GA in cells through transcriptome analysis (Figure 1). For this, GA-induced differential GEPs were first obtained from each cell line (A549, PC3, and MCF7). To investigate the potential biological functions of the DEGs, Gene Ontology (GO) [23] and the Kyoto Encyclopedia of Genes and Genomes (KEGG) pathway [24] enrichment analyses were performed. To further specify the direct targets and mechanisms, GA-induced GEP changes were compared with those induced by FDA-approved drugs in the LINCS L1000 database, and the drugs that exhibited high similarities in expression patterns in the three cell lines were identified. Connectivity scores were used to quantify the similarity in GEP changes, and the list was compiled in descending order based on the connectivity scores. The top 10 FDA-approved drugs on the list were identified. Then, their respective targets for inhibition and antagonism were elucidated, yielding a total of 19 potential targets. Based on the high similarity in GEPs, we hypothesized that GA could inhibit the same target proteins as these drugs. Using this comparative transcriptome analysis, we aimed to identify the multiple mechanisms used by GA and how changes in transcriptional activity induced by GA may reflect or contribute to diseases. As a small molecule, GA exhibits diverse mechanisms in different tissues; hence, more comprehensive research at the genome level is warranted.

## 2. Results

### 2.1. Data Analysis of GA-Induced GEPs

To obtain the GEPs, we treated three cell lines (A549 (lung carcinoma epithelial cell line), PC3 (human prostate adenocarcinoma cell line), and MCF7 (human breast carcinoma cell line)) with 10 µM GA for 6 h; DMSO served as the control. We calculated the mean value for triplicate samples individually and obtained the fold change (FC) by dividing the gene expression value of the GA treatment group by that of the control group, where FC represented the GEP changes in each cell line. An absolute value of FC < 1 indicated that gene expression was downregulated after GA treatment, which was expressed using a negative (−) sign.

For differentially expressed gene (DEG) analysis, *p*-values of < 0.05 indicated significant changes in the gene expressions, and the FC cutoff value was set at FC ≥ 1.5. In A549 cells, there were 76 upregulated genes and 60 downregulated genes. In PC3 cells, 54 and 62 genes were upregulated and downregulated, respectively, whereas in MCF7 cells, 55 and 50 genes were upregulated and downregulated, respectively. Among these genes, pogo transposable element derived with ZNF domain (*POGZ*) and T cell receptor-associated transmembrane adaptor 1 (*TRAT1*) were downregulated in both A549 and MCF7 cell lines, whereas Formin 1 (*FMN1*) was downregulated in both PC3 and MCF7 cell lines. Notably, no common regulated genes were identified in all three cell lines (Figure 2 and Appendix A). *TRAT1* regulates T cell-mediated immune responses [25], while *POGZ* is involved in neurodevelopment and mitotic cell cycle progression [26], and *FMN1* regulates the organization of the actin cytoskeleton [27], which is key to many cellular processes, including the cell cycle, motility, division, and adhesion. All three genes (i.e., *TRAT1*, *POGZ*, and *FMN1*) have been identified as potential therapeutic targets for the treatment of some cancers [25,28,29].

### 2.2. DEG Analysis through GO Mapping and KEGG Pathway Enrichment

We performed GO and KEGG pathway enrichment analyses using the Database for Annotation, Visualization, and Integrated Discovery (DAVID) [30,31] (Appendix A). Among the three GO sub-ontologies (biological process [BP], cellular component, and molecular function), we focused on BP to assess the underlying biological features. First, we extracted the GO Biological Process (GOBP) terms that were significantly enriched (*p* < 0.05) in each cell line. To further examine the regulatory mechanisms, the gene lists from the GOBP terms of each cell line were merged, resulting in 21 upregulated and 30 downregulated gene sets. Thereafter, each gene set was subjected to GO enrichment analysis (Figure 3). For the merged gene set comprising upregulated genes, the top-ranked GOBP terms were “regulation of transcription from RNA polymerase II promoter” and “positive regulation of transcription from RNA polymerase II promoter,” whereas the top GOBP terms for the merged gene set comprising downregulated genes included “signal transduction,” “positive regulation of transcription from RNA polymerase II promoter,” and “intracellular signal transduction” (Appendix A).

In KEGG pathway enrichment analysis, we did not identify any significantly enriched pathways for the upregulated gene set of three cell lines; however, 12 genes were identified from significantly enriched metabolic pathways for the downregulated gene set of three cell lines (*p* < 0.05) (Figure 4). The top-ranked pathways for these downregulated genes included pathways of neurodegeneration—multiple diseases, pathways in cancer, dopaminergic synapse, axon guidance, lipid and atherosclerosis, Parkinson’s disease, and Huntington’s disease (Appendix A).

### 2.3. LINCS L1000 Data-Based Expression Pattern Analysis

To better understand the molecular pathways affected by GA, we used a computational drug screening platform to discover the potential target proteins that directly interact with GA. In our previous studies, we used a computational program to search for FDA-approved drugs with the highest similarity in GEP patterns [32]. Using this method, we compared the GEP changes induced by GA with those induced by the drugs in the LINCS L1000 library. Intergroup similarity was measured using the connectivity score method introduced by Zhang et al. [33]. Table 1 summarizes the top 10 drugs with the highest similarity based on the average Zhang score of the three cell lines, and their 2D structures are shown in Figure 5. We mapped the LINCS perturbagen IDs to the corresponding approved drug names using DrugBank IDs [34]. Because GA shares target proteins with these drugs due to their highly similar expression patterns, we also compiled the known target proteins of each drug (Table 1).

We focused on inhibitor drugs as they are comparatively widely available and easier to develop than activator drugs. We identified 19 antagonists and/or inhibitor proteins from DrugBank as the targets of the top 10 drugs (Table 1).

To elucidate the functions of the 19 target proteins, GO and KEGG pathway enrichment analyses were performed. The top-ranked GOBP terms (*p* < 0.05) were regulation of synaptic vesicle exocytosis, DNA replication, DNA topological change, response to drugs, and response to xenobiotic stimuli (Figure 6a and Appendix A). KEGG pathway enrichment analysis (*p* < 0.05) indicated significant enrichment in the cAMP signaling, calcium signaling, neuroactive ligand-receptor interaction, and cGMP-PKG signaling pathways (Figure 6b and Appendix A).

## 3. Discussion

In this study, we investigated the molecular mechanisms that GA may affect by conducting transcriptomic analysis based on the GEPs before and after GA treatment. We examined the GEP changes in three cancer cell lines before and after GA treatment using microarray analysis. Pathway enrichment analysis using DEGs revealed that GA downregulated the expression of genes related to neurodegenerative pathways, such as Parkinson’s and Huntington’s disease, as well as those associated with the nervous system, such as dopaminergic synapse and axon guidance.

However, this transcriptome-based pathway enrichment analysis mainly highlights the regulatory mechanisms at the pathway level, with limited information on precise mechanisms or the direct identification of inhibitory targets. To overcome these limitations, we adopted an alternative approach to compare the microarray experimental data of GA with that of the drugs in the LINCS L1000 database. This database offers information on the GEPs of various cell lines, which were induced by thousands of different perturbagens, including FDA-approved drugs with known action mechanisms. By comparing the GEP changes induced by GA with those induced by the drugs in the database, we identified the drugs showing similar GEP changes. A higher similarity indicated a higher likelihood that GA directly or indirectly targeted the known target proteins of the identified drugs.

In this study, we employed a computational program used in earlier studies to identify the drugs exhibiting similar GEP patterns. The highly accurate Zhang score, a connectivity score, was used to measure the similarity between the two GEP datasets [33]. The program allowed us to sort drugs in descending order of the Zhang score, enabling us to easily specify the top 10 drugs that induce the most similar changes in GEP as those triggered by GA. Interestingly, we observed that even with significant differences in molecular size, GA was able to induce changes in gene expression patterns similar to those induced by drugs. This may be due to the multitarget effects. However, there are many challenges in achieving a full understanding of the underlying mechanisms due to biological complexity.

Using the top 10 drug lists, we extracted drug-target protein interaction information from DrugBank [34] and identified the proteins that act as inhibitors. Overall, this approach proved to be a convenient bioinformatics tool for comparative analyses of multiple experimental groups to identify differences between them. By identifying the approved drugs with high similarity in GEP changes using LINCS L1000, we could specify the potentially shared targets of GA. Because this dataset covers most of the genes expressed in cells (widely expressed landmark genes that are directly measured as well as other genes that could be inferred), it effectively served as a useful analytical method for our study.

Using the gene list of the specified target proteins, GO and KEGG pathway enrichment analyses were performed. The results revealed the GOBP terms associated with the nervous system, such as regulation of synaptic vesicle exocytosis. In KEGG pathway analysis, cAMP signaling, calcium signaling, and neuroactive ligand–receptor interaction were among the selected pathways.

We considered these proteins as potential targets of GA and found that GA may directly or indirectly affect the related regulatory pathways of 10 drugs and their target proteins. For instance, cytarabine and fludarabine bind directly to DNA polymerase β and ribonucleoside-diphosphate reductase, respectively. GA is reported to directly bind to these proteins, thus exerting anti-diabetic effects [35]. Additionally, there is evidence suggesting that GA downregulates USP47, which is known to stabilize DNA polymerase β, potentially disabling base excision repair (BER) [36]. Also, GA has been reported to induce p53 activation in non-small-cell lung cancer (NSCLC) cells, leading to the suppression of cancer cell survival and exhibiting a tumor-suppressive effect [37]. In terms of the pathway-based approach, these findings suggest the possibility that GA may be involved in pathways associated with the potential target proteins. Sirolimus targets the serine/threonine-protein kinase: mammalian target of rapamycin (mTOR), and GA has been associated with the Akt/mTOR signaling pathway, demonstrating antileukemic efficacy in acute myeloid leukemia (AML) [38]. Cyclosporine directly binds calcineurin, inhibiting T cell activation by preventing nuclear factor of activated T cells (NF-AT) activation. It has been reported that GA may suppress cardiac hypertrophic remodeling and heart failure through inhibition of calcineurin and NF-AT in cardiac cells [39]. In a Parkinson’s disease model, GA exhibits neuroprotective effects through oxidative stress induction [40], with potential associations with muscarinic receptors implicated in oxidative stress regulation [41]. This finding suggests a possible connection between GA and muscarinic acetylcholine receptors, one of the target proteins of profenamine. Another known target protein of profenamine, NMDA glutamate receptors, may also be associated with GA in the context of glutamate-induced neurotoxicity and neuroprotective effects [42]. Trifluoperazine directly blocks dopamine D1, D2 receptors, and alpha-1A adrenergic receptors. In a mouse model, the antidepressant-like effect of GA is inhibited when treated with antagonists of dopamine D2 and alpha-1A adrenergic receptors, suggesting a potential connection with the dopaminergic and adrenergic pathways [43]. Trifluoperazine also targets calmodulin, and it has been reported that GA attenuates calcium-calmodulin-dependent kinase II-induced apoptosis in cardiac cells [44]. The known targets of daunorubicin, doxorubicin, and topotecan are DNA topoisomerases for their anti-tumor effects. GA has been reported to indirectly stabilize DNA topoisomerase I- and II-DNA complexes through hydrogen peroxide generation, inducing apoptosis [45]. Finally, digoxin is known to inhibit Na+, K+-ATPase, and a phytochemical study shows that *Cuphea glutinosa*, containing a high fraction of GA, inhibits Na+, K+-ATPase activity [46], suggesting a potential association between GA and Na+, K+-ATPase. To identify whether there are any interconnections between the target proteins, we performed the protein-protein interaction (PPI) analysis (Appendix A). While there seem to be certain connections between the target proteins, these connections appear to arise from the similarity within targets for a single drug, and there appears to be no distinct specialization in their interactions.

This suggests that our results using LINCS L1000 align with previous studies on GA, and support the idea of considering these proteins as potential direct targets. These findings highlight the potential of this transcriptome analysis method not only for GA but also for investigating the molecular mechanisms of various natural molecules. Nonetheless, since the results of this study are predicted using a computational program, further experimental validation is warranted.

While numerous studies have been conducted to explore phytochemicals as crucial sources in drug discovery, there have been relatively few investigations focusing on transcriptomic profile changes induced by phytochemicals. This study aims to propose a methodology for investigating natural compounds with multifaceted and comprehensive therapeutic effects, such as GA. GA, a polyphenol compound found in various plants, is renowned for its potent antioxidant and anti-inflammatory properties, contributing to therapeutic activities in cardiovascular diseases, cancer, neurodegenerative disorders, and aging [1]. Consequently, GA and its derivatives are often used as promising lead compounds for new drug development, contributing to drug modeling and medicinal chemistry research [47,48]. However, research on the mechanisms of GA has been limited due to its complexity. Recently, useful data sources, such as the LINCS database, became available for interpreting mechanisms of action, allowing us to conduct this study. The strategy proposed in this study is expected to be applicable to the investigation of the molecular mechanisms of these natural small compounds. Our study examines changes in transcriptome expression patterns, proposing potential targets for the action of GA. Based on these findings, future studies may clarify the regulatory mechanisms of GA or explore various phytochemicals, such as GA, using the methodological strategies employed in this study. Our findings may provide valuable clues not only for understanding the action mechanisms of phytochemicals but also for identifying potential side effects.

## 4. Materials and Methods

### 4.1. Materials

GA (CAS Registry No. 149-91-7; 3,4,5-trihydroxybenzoic acid) was purchased from Shaanxi Sciphar Natural Products Co., Ltd. (Shangluo, Shaanxi, China). The three cancer cell lines, namely lung carcinoma cells (A549), human prostate adenocarcinoma cells (PC3), and human breast carcinoma cells (MCF7), were purchased from the American Type Culture Collection (ATCC, Rockville, MD, USA).

### 4.2. Cell Culture

A549 and PC3 cells were cultured in RPMI-1640 supplemented with 10% fetal bovine serum (FBS) and 1% penicillin-streptomycin-glutamine. To culture PC3 cells, 1 mM sodium pyruvate and 10 mM 4-(2-hydroxyethyl)-1-piperazineethanesulfonic acid were additionally added to the media. MCF7 cells were cultured in Dulbecco’s modified Eagle’s medium supplemented with 10% FBS and 1% penicillin-streptomycin-glutamine. All cells were cultured at 37 °C in a 5% CO_2_ humidified atmosphere and incubated for 2 weeks after initial seeding for stabilization.

### 4.3. RNA Sample Preparation

GA was dissolved in 10 mM DMSO and stored at −80 °C. Each cell line was plated in six 60-mm dishes (three each for DMSO and GA treatments). The cell density was 1.3 × 10^6^ cells for A549, 1.496 × 10^6^ cells for PC3, and 8.45 × 10^5^ cells for MCF7. After 24 h, the cells were treated with 1000-fold diluted DMSO and GA, resulting in final concentrations of 0.1% for DMSO and 10 μM for GA. After 6 h of DMSO and GA treatment, RNA was extracted using the RNeasy Mini Kit (Qiagen, Hilden, Germany) according to the manufacturer’s instructions. The extracted RNAs were stored at −80 °C.

### 4.4. Microarray Data Analysis

Total RNA samples were sent to Macrogen (Seoul, Korea) for assessment using the Clariom™ S Assay, Human. The ND-2000 spectrophotometer (NanoDrop, Wilmington, NC, USA) was used to detect RNA purity, and the Agilent 2100 bioanalyzer (Agilent Technologies, Palo Alto, CA, USA) was used to detect RNA integrity.

For the Affymetrix whole-transcript (WT) expression array process, the GeneChip WT PLUS Reagent Kit (Affymetrix, Santa Clara, CA, USA) was used according to the manufacturer’s instructions. cDNA was synthesized using the GeneChip WT Amplification Kit (Affymetrix) according to the manufacturer’s instructions.

Thereafter, sense cDNA was fragmented and labeled with biotin using terminal deoxynucleotidyl transferase using the GeneChip WT Terminal Labeling Kit (Affymetrix). Approximately 5.5 μg of labeled DNA target was incubated at 45 °C for hybridization with the Affymetrix GeneChip Human Clariom S Array for 16 h. After washing and staining on the GeneChip Fluidics Station 450, the hybridized arrays were scanned on the GCS3000 Scanner (Affymetrix). The Affymetrix^®^ GeneChip™ Command Console software was used to calculate the signal values.

Gene enrichment, pathway, and functional annotation analyses were performed to obtain a probe list using the DAVID functional annotation tool [30,31].

### 4.5. Identification of the Potential Target Proteins Using the LINCS L1000 Database

The LINCS L1000 database was used to screen the approved drugs that generate GEP changes similar to GA, among thousands of small molecules, including FDA-approved drugs in various cell lines at different time points and doses. The LINCS L1000 level 5 datasets that were used in this study are available in Gene Expression Omnibus (Accession number GSE92742, https://www.ncbi.nlm.nih.gov/geo/query/acc.cgi?acc=GSE92742, accessed on 18 January 2023). This dataset contains information on the GEPs of 978 landmark genes, collected from the L1000 assay [16]. The raw data obtained from LINCS L1000 were filtered based on cell type and gene expression. Note that the conditions selected for this step were similar to those for cell treatment at a dose of 10 μM GA for 6 h. FC was measured by dividing the mean expression under GA by that under DMSO for each cell line. The obtained FC was normalized using log2 and used to compare with that of the 978 landmark GEPs in the LINCS database.

A computational program that was used in one of our previous studies was used to calculate the similarity in GEP patterns between GA and the approved drugs using connectivity scores and identify the approved drugs that yielded changes most similar to those induced by GA [32]. Briefly, DrugBank was used to obtain the approved drug list from the LINCS database, and connectivity scores (the Zhang score) were used to identify the approved drugs that induce GEP changes similar to those induced by GA. As a comparative method, Lamb et al. developed CMap16 by creating a database using the GEP patterns of known chemicals and generating ranks for the test DEGs based on their expression level [15]. Subsequently, Zhang et al., introduced an improved approach to compute the connectivity scores, offering a more simplified and precise method for comparing GEP changes between groups [33]. In this method, a query signature (GA treatment) is compared to each of the drug-gene expression profiles ranked in the LINCS L1000 database based on their respective scores. The score ranges from −1 (representing the highest negative correlation) to 1 (representing the highest positive correlation), implying that a higher score indicates greater similarity. Target proteins were restricted to antagonists or inhibitors with pharmacological activity. All program scripts used in this computational software can be found in the online repository (https://bitbucket.org/krictai/predms/src/master/, accessed on 19 February 2023).

## 5. Conclusions

In this study, we investigated the underlying mechanisms by which GA affects cells using transcriptome-based comparative analysis. Through transcriptome analysis, potential target proteins for GA were identified. We used the LINCS L1000 database to compare the changes in GEPs induced by GA with those induced by approved drugs and identify the drugs with similar GEPs. This was crucial between the GEPs of approved drugs, and GA may indicate a similar mechanism of action. Nineteen potential target proteins were obtained, consistent with the findings of previous studies, providing further support and validation. In particular, our study indicates the direct binding of GA to DNA polymerase β and ribonucleoside-diphosphate reductase, both of which were among the top target proteins. Consequently, it is probable that GA may play a role in base excision DNA repair pathways and cell survival within p53-dependent cell survival pathways, as discussed earlier. Our findings highlight the potential of transcriptome-based comparative analysis in elucidating the molecular mechanisms of small natural compounds. To address the complex biological challenges of small compounds such as GA, a genome-wide approach using transcriptome profiles can be a powerful method to gain a comprehensive view of their multifaceted actions. The availability of valuable data sources, such as the LINCS database, has recently enabled such a study, proposing a new strategy that may be applicable to the investigation of the molecular mechanisms of these natural small compounds. Nevertheless, additional studies are warranted to comprehensively elucidate the specific targets and mechanisms of GA. This can help assess the action mechanisms of other natural compounds with therapeutic potential.

## Figures and Tables

**Figure 1 ijms-25-02303-f001:**
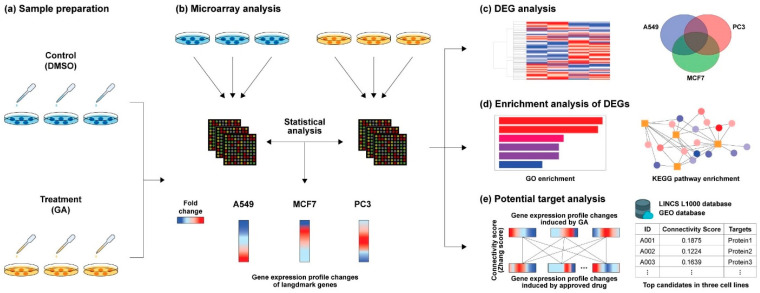
Overall study scheme (**a**) gallic acid (GA) or dimethyl sulfoxide (DMSO) was used to treat three cell lines for sample preparation. (**b**) Transcriptome profiles of each cell line were obtained through microarray analysis, and the fold change was calculated to identify differentially expressed genes (DEGs). (**c**) DEGs were listed for each cell line, and commonly expressed genes in the three cell lines were identified. (**d**) For DEGs, Gene Ontology (GO) and Kyoto Encyclopedia of Genes and Genomes (KEGG) pathway enrichment analyses were conducted. (**e**) The gene expression profile (GEP) changes were compared with those of FDA-approved drugs in LINCS L1000 by calculating the average connectivity scores (Zhang score) using computational software. The higher the score, the greater the similarity in GEPs. The known target proteins of the top 10 drugs were identified for further analysis.

**Figure 2 ijms-25-02303-f002:**
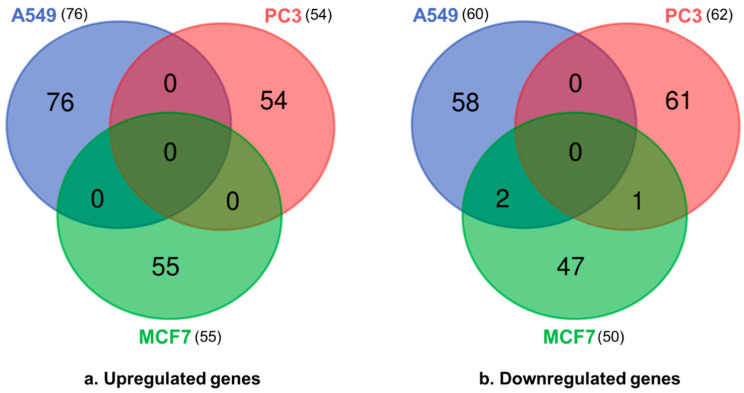
Venn diagrams for the (**a**) upregulated and (**b**) downregulated differentially expressed genes (DEGs) in the three cell lines (A549, PC3, and MCF7). Overlapping regions represent DEGs shared in all three groups. The cutoff for statistically significant DEGs was fold change ≥ 1.5 and *p*-value < 0.05. The Venn diagrams were created using the UGent website (https://bioinformatics.psb.ugent.be/webtools/Venn/, accessed on 20 March 2023).

**Figure 3 ijms-25-02303-f003:**
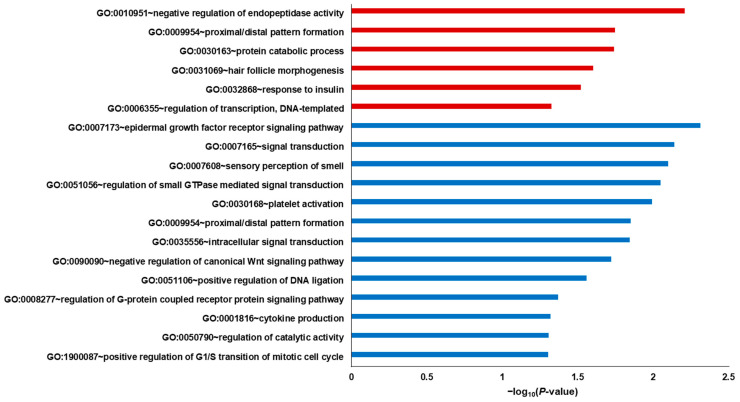
Gene Ontology Biological Process (GOBP) enrichment analysis results of the merged gene sets. Red bars reflect the results of the upregulated gene set, and blue bars represent the results of the downregulated gene set.

**Figure 4 ijms-25-02303-f004:**
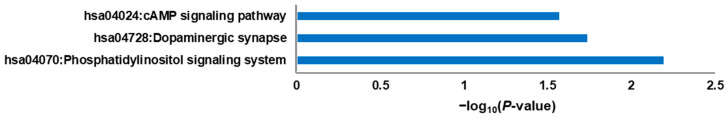
Kyoto Encyclopedia of Genes and Genomes (KEGG) pathway enrichment analysis results of the merged gene set. Blue bars represent the results of the downregulated gene set.

**Figure 5 ijms-25-02303-f005:**
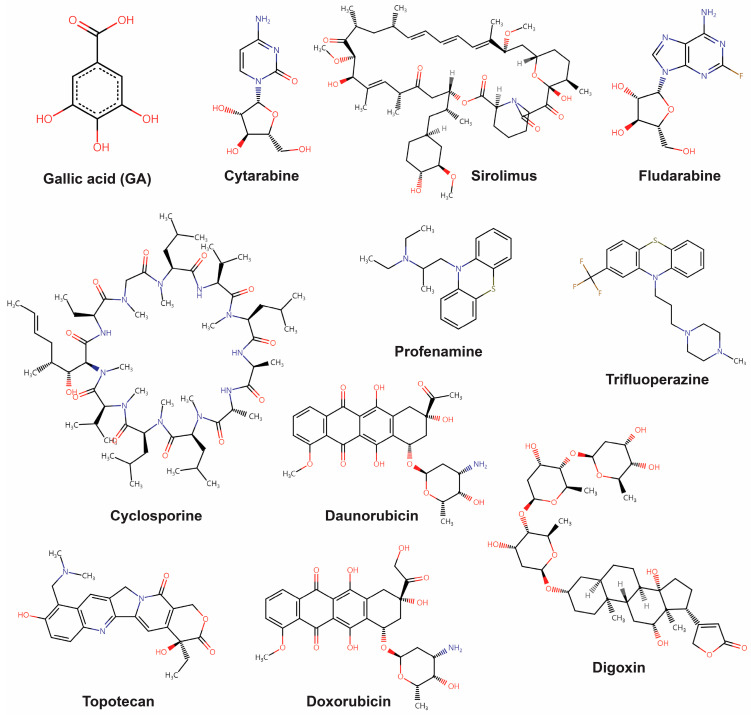
Structures of GA and the top 10 drugs.

**Figure 6 ijms-25-02303-f006:**
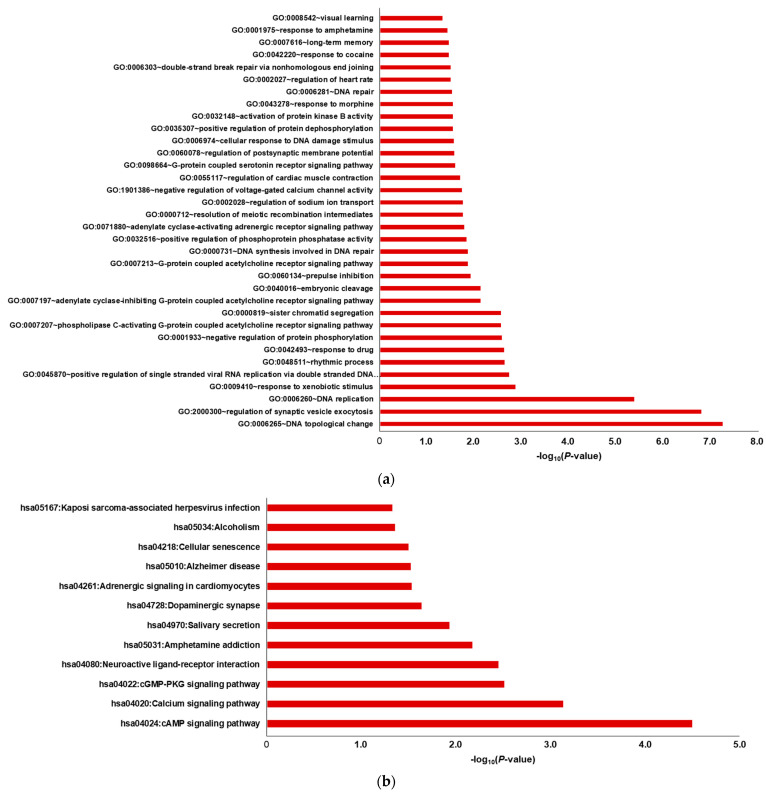
Results of the enriched (**a**) GOBP terms and (**b**) KEGG pathways of the potential target genes. The red bars represent the results of the upregulated gene set.

**Table 1 ijms-25-02303-t001:** Top 10 drug lists from the LINCS L1000 dataset with the most similar GEP patterns.

	Perturbagen ID	Drug Name	Mean of the Zhang Score	Targets		
				Gene Symbol	UniProt ID	Name of the Target Proteins
1	BRD-K33106058_DB00987	Cytarabine	0.402339	*POLB*	P06746	DNA polymerase beta
2	BRD-K89626439_DB00877	Sirolimus (rapamycin)	0.396491	*MTOR*	P42345	Serine/threonine-protein kinase mTOR
3	BRD-K72238567_DB01073	Fludarabine	0.20117	*POLA1*, *RRM1*	P09884, P23921	DNA polymerase alpha catalytic subunit Ribonucleoside-diphosphate reductase large subunit
4	BRD-A38030642_DB00091	Cyclosporine	0.192982	*PPP3R2*, *PPIA*	Q96LZ3, P62937	Calcineurin subunit B type 2 Peptidyl-prolyl cis-trans isomerase A
5	BRD-A16311756_DB00392	Profenamine	0.166082	*CHRM1*, *CHRM2*, *GRIN3A*	P11229, P08172, Q8TCU5	Muscarinic acetylcholine receptor M1 Muscarinic acetylcholine receptor M2 Glutamate receptor ionotropic, NMDA 3A
6	BRD-K89732114_DB00831	Trifluoperazine	0.153216	*DRD2*, *CALY*, *ADRA1A*, *CALM*, *S100A4*	P14416, Q9NYX4, P35348, P0DP23, P26447	Dopamine D2 receptor Neuron-specific vesicular protein calcyon Alpha-1A adrenergic receptor Calmodulin Protein S100-A4
7	BRD-K43389675_DB00694	Daunorubicin	0.145029	*TOP2A*, *TOP2B*	P11388, Q02880	DNA topoisomerase 2-alpha DNA topoisomerase 2-beta
8	BRD-K23478508_DB00390	Digoxin	0.138012	*ATP1A1*	P05023	Sodium/potassium-transporting ATPase subunit alpha-1
9	BRD-A59985574_DB01030	Topotecan	0.127485	*TOP1*, *TOP1MT*	P11387, Q969P6	DNA topoisomerase 1 DNA topoisomerase I, mitochondrial
10	BRD-K04548931_DB00997	Doxorubicin	0.100585	*TOP1*, *TOP2A*, *TOP2B*	P11387, P11388, Q02880	DNA topoisomerase 1 DNA topoisomerase 2-alpha DNA topoisomerase 2-beta

The Zhang score (connectivity score) was used as a measure of similarity. A higher score indicates greater similarity. Target proteins were restricted to antagonists or inhibitors with pharmacological activity.

## Data Availability

All related data are present within the manuscript.

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
