# Peer review of "Investigation of the General Molecular Mechanisms of Gallic Acid via Analyses of Its Transcriptome Profile"

_ijms, 2024, doi:10.3390/ijms25042303_

Round 1
Reviewer 1 Report
Comments and Suggestions for Authors
Kim et al studied the “Investigation of the Pharmacological Mechanisms of Gallic 2 Acid by Analyzing Transcriptome Profile Changes”. The study seems interesting but there are still some issues could be raised as following questions:
1. Title seems nonsuitable-Pharmacological mechanisms for which pathogenesis?
2. Do the three cell lines represent all the biological functions? If not, how the changes of gene expression could be compared in context of these three? Importantly, no pathogenesis is pronounced.
3. It’s a multitarget analyses, right? If so, comprehension of GA interaction with multiple genes and proteins are involved?
4. Authors did not show the Interaction plot of proteins (PPI) of common interacting genes utilizing the degree algorithm.
5. The representing nodes of potent proteins with interconnections (edges) between them is missing for their multi-pathway involvement, and
6. hierarchy tree of pathways is missing.
7. Impact on PPI Network Analysis of Common Targets should be addressed using STRING database
8. Through pertinent targets’ molecular function (MF), biological process (BP) and cellular components (CC) groups are not clearly identified.
9. The complete pathview needs to be designed for the GA action
10. Line 289-the access date to the link needs to be mentioned.
11. In the conclusion-Very specific protein-based pathway needs to be consized.
12. The references should not be older than 10 years except there is an inevitable method or protocol.
Comments on the Quality of English Language
Please see the attached file
Author Response
- Title seems nonsuitable-Pharmacological mechanisms for which pathogenesis?
[Response] Thank you for raising this concern. We agree with the reviewer's comments. Following the reviewer's suggestion, we have revised the title of the manuscript to ‘Investigation of the General Molecular Mechanisms of Gallic Acid via Analyses of its Transcriptome Profile Changes.’
- Do the three cell lines represent all the biological functions? If not, how the changes of gene expression could be compared in context of these three? Importantly, no pathogenesis is pronounced.
[Response] Thank you for your comments. We agree with your comments. However, the purpose of analyzing the three cell lines used in this study was not to analyze the biological functions of GA. Instead, the purpose was to examine the patterns of gene expression and investigate the molecular mechanisms (i.e., molecular targets) through which GA functions and comparing it with drugs whose molecular mechanisms of action are known. We believe that the title may have caused some confusion. To make it clearer, we have revised the title and made minor revisions in the abstract and main text (Page 1, line 14; Page 7, line 180)
3.It’s a multitarget analyses, right? If so, comprehension of GA interaction with multiple genes and proteins are involved?
[Response] Thank you for these valuable questions. Your assumption is correct. Therefore, we conducted a comparative analysis of the transcriptome profiles altered by GA and those of drugs with known mechanisms of action. We selected the top 19 proteins that are believed to be regulated by GA (Table 1).
4, 5, 6. Authors did not show the Interaction plot of proteins (PPI) of common interacting genes utilizing the degree algorithm. The representing nodes of potent proteins with interconnections (edges) between them is missing for their multi-pathway involvement, and hierarchy tree of pathways is missing.
[Response] Thank you for pointing this out. The results of the protein-protein interaction (PPI) analysis are presented in the figure below. While there seem to be certain connections between the target proteins, these connections appear to arise from the similarity in targets for a single drug, and there appears to be no distinct specialization in their interactions. We have included this figure as Figure S1 in the Supplementary Materials and mentioned it in the main text (Page 9, line 255).
- Impact on PPI Network Analysis of Common Targets should be addressed using STRING database
[Response] We appreciate your suggestions. In response to question 4, we have used the STRING database for providing the answer.
- Through pertinent targets’ molecular function (MF), biological process (BP) and cellular components (CC) groups are not clearly identified.
[Response] Thank you for raising this concern. We have now added a table in Supplementary Materials (Table S6).
- The complete pathview needs to be designed for the GA action
[Response] Thank you for your invaluable suggestions. Although enriched pathways among proteins exist, as mentioned earlier, there does not seem to be a significant specialization or specific interaction among proteins in particular pathways. Additionally, due to the lack of clarity regarding GA's targets, drawing a comprehensive pathway view for a single pathway may not provide meaningful insights. Therefore, we have included the results of KEGG pathway enrichment in Figure 4 and added detailed information in Table S7 (Supplementary Materials).
- Line 289-the access date to the link needs to be mentioned.
[Response] Thank you for pointing out this issue. The link contained empty spaces, causing it to malfunction. We have now removed the spaces and, as per your suggestions, added the date of accession (Accession number GSE92742, https://www.ncbi.nlm.nih.gov/geo/query/acc.cgi?acc=GSE92742, Jan 2023).
- In the conclusion-Very specific protein-based pathway needs to be consized.
[Response] Thank you for the insightful suggestion. As per your suggestion, we have added detailed explanations regarding the protein-based pathways in terms of GA regulatory mechanisms as follows (Page 8, line 225; Page 10, line 365):
“Additionally, there is evidence suggesting that GA downregulates USP47, which is known to stabilize DNA polymerase β, potentially disabling BER [36]. Also, GA has been reported to induce p53 activation in non-small-cell lung cancer (NSCLC) cells, leading to the suppression of cancer cell survival and exhibiting a tumor-suppressive effect [37]. In terms of the pathway-based approach, these findings suggest the possibility that GA might be involved in pathways associated with the potential target proteins.”
“In particular, our study indicates the direct binding of GA to DNA polymerase β and ribonucleoside-diphosphate reductase, both of which were among the top target proteins. Consequently, it is probable that GA may play a role in base excision DNA repair pathways and cell survival within p53-dependent cell survival pathways, as discussed earlier.”
- The references should not be older than 10 years except there is an inevitable method or protocol.
[Response] Thank you for pointing this out. We have modified some old references except the ones that are inevitable.
Reviewer 2 Report
Comments and Suggestions for Authors
I noticed the fact that I could not visualize the additional material, asking the authors of the manuscript to clarify the inconvenience.
Next, I prepared some suggestions for the authors of the manuscript, in order to improve its quality.
Suggestions for the authors of the manuscript, in order to improve the article:
1. I suggest you specify what arguments there are in explaining the similarity of gallic acid, a small molecule compared to the drugs discussed, which are mostly macrolides.
2. Please explain, what are the 19 potential targets identified
3. I suggest you write the structural formulas of the 10 drugs and gallic acid from table 1, as they are necessary
4. I suggest you describe the therapeutic effect of the 10 drugs, in the idea of similarity with the potential therapeutic effect of gallic acid.
5. I suggest that he explain what the elements of originality are.
6. To explain how to evaluate the Zhang score, mentioned in the text or as the legend to table 1.
7. We found in the review of the manuscript that there is an error in the view of the Supplementary Materials, suggesting that you resolve this inconvenience.
8. Regarding the conclusions, it would be indicated to certify in more detail that gallic acid can be a model to support the elucidation of the action mechanisms of other natural compounds with therapeutic potential.
Author Response
I noticed the fact that I could not visualize the additional material, asking the authors of the manuscript to clarify the inconvenience.
Next, I prepared some suggestions for the authors of the manuscript, in order to improve its quality.
Suggestions for the authors of the manuscript, in order to improve the article:
1. I suggest you specify what arguments there are in explaining the similarity of gallic acid, a small molecule compared to the drugs discussed, which are mostly macrolides.
[Response] Thank you for the suggestion. We agree with your concerns. A key objective of our study was to analyze the molecular mechanisms of the small molecule GA. Interestingly, we observed that even with significant differences in molecular size, GA was able to induce changes in gene expression patterns similar to those induced by drugs, possibly due to the multitarget effects. However, there are many challenges in achieving a full understanding of the underlying mechanisms due to the biological complexity. We have added certain information in response to your comment in the main text (Page 7, line 203).
Please explain, what are the 19 potential targets identified
[Response] Table 1 shows the list of the 19 potential target proteins. Excluding duplicate names, the remaining proteins are: DNA polymerase beta, Serine/threonine-protein kinase mTOR, DNA polymerase alpha catalytic subunit, ribonucleoside-diphosphate reductase large subunit, calcineurin subunit B type 2, peptidyl-prolyl cis-trans isomerase A, muscarinic acetylcholine receptor M1, muscarinic acetylcholine receptor M2, glutamate receptor ionotropic (NMDA 3A), dopamine D2 receptor, neuron-specific vesicular protein calcyon, alpha-1A adrenergic receptor, calmodulin, protein S100-A4, DNA topoisomerase 2-alpha, DNA topoisomerase 2-beta, sodium/potassium-transporting ATPase subunit alpha-1, DNA topoisomerase 1, and DNA topoisomerase I (mitochondrial).
I suggest you write the structural formulas of the 10 drugs and gallic acid from table 1, as they are necessary
[Response] Thank you for your kind suggestions. As per your suggestions, we have added a figure showing the 2D structures of GA and the 10 drugs (Figure 5).
Figure 5.
- I suggest you describe the therapeutic effect of the 10 drugs, in the idea of similarity with the potential therapeutic effect of gallic acid.
[Response] We appreciate your insightful suggestions. As suggested, descriptions of the therapeutic effects of the 10 drugs in this study have been added to the main text in context of the similarity between the 10 drugs and GA (Page 8, line 221).
“We considered these proteins as the potential targets of GA and found that GA may directly or indirectly affect the related regulatory pathways of 10 drugs and their target proteins. For instance, cytarabine and fludarabine bind directly to DNA polymerase β and ribonucleoside-diphosphate reductase, respectively. GA is reported to directly bind to these proteins, playing a role in its anti-diabetic effects [35]. Additionally, there is evidence suggesting that GA downregulates USP47, which is known to stabilize DNA polymerase β, potentially disabling base excision repair [36]. Also, GA has been reported to induce p53 activation in non-small-cell lung cancer (NSCLC) cells, leading to the suppression of cancer cell survival and exhibiting a tumor-suppressive effect [37]. In terms of the pathway-based approach, this suggests the possibility that GA may be involved in pathways associated with the potential target proteins. Sirolimus targets serine/threonine-protein kinase: mammalian target of rapamycin (mTOR), and GA has been associated with the Akt/mTOR signaling pathway, demonstrating antileukemic efficacy in acute myeloid leukemia (AML) [38]. Cyclosporine directly binds calcineurin, inhibiting T cell activation by preventing nuclear factor of activated T-cells (NF-AT) activation. It has been reported that GA may suppress cardiac hypertrophic remodeling and heart failure through inhibition of calcineurin and NF-AT in cardiac cells [39]. In a Parkinson’s disease model, GA exhibits neuroprotective effects through oxidative stress induction [40], with potential associations with muscarinic receptors implicated in oxidative stress regulation [41]. This finding suggests a possible connection between GA and muscarinic acetylcholine receptors, one of the target proteins of profenamine. Another known target protein of profenamine, NMDA glutamate receptors, may also be associated with GA in the context of glutamate-induced neurotoxicity and neuroprotective effects [42]. Trifluoperazine directly blocks dopamine D1, D2 receptors, and alpha-1A adrenergic receptors. In a mouse model, the antidepressant-like effect of GA is inhibited when treated with antagonists of dopamine D2 and alpha-1A adrenergic receptors, suggesting a potential connection with the dopaminergic and adrenergic pathways [43]. Trifluoperazine also targets calmodulin, and it has been reported that GA attenuates calcium calmodulin-dependent kinase II-induced apoptosis in cardiac cells [44]. The known targets of daunorubicin, doxorubicin, and topotecan are DNA topoisomerases for their anti-tumor effects. GA has been reported to indirectly stabilize DNA topoisomerase I- and II-DNA complexes through hydrogen peroxide generation, inducing apoptosis [45]. Finally, digoxin is known to inhibit Na+, K+-ATPase, and a phytochemical study shows that Cuphea glutinosa, containing a high fraction of GA, inhibits Na+, K+-ATPase activity [46], suggesting a potential association between GA and Na+, K+-ATPase. To identify whether there are any interconnections among the target proteins, we performed the protein-protein interaction (PPI) analysis (Figure S1). While there seems to be some connections between the target proteins, these connections appear to arise from the similarity within targets for a single drug, and there appears to be no distinct specialization in their interactions.”
I suggest that he explain what the elements of originality are.
[Response] Thank you for your valuable recommendation. In response to your comment, we have added a paragraph in the main text as following (Page 8, line 266):
“While numerous studies have been conducted to explore phytochemicals as crucial sources in drug discovery, there have been relatively little investigations focusing on transcriptomic profile changes induced by phytochemicals. This study aims to propose a methodology for investigating natural compounds with the multifaceted and comprehensive therapeutic effects such as GA. GA, a polyphenol compound found in various plants, is renowned for its potent antioxidant and anti-inflammatory properties, contributing to therapeutic activities in cardiovascular diseases, cancer, neurodegenerative disorders, and aging [1]. Consequently, GA and its derivatives are often used as promising lead compounds for new drug development, contributing to drug modeling and medicinal chemistry research [47,48]. However, research on the mechanisms of GA has been limited due to its complexity. Recently, the useful data sources, such as the LINCS database, became available for interpreting mechanisms of action, allowing us to con-duct this study. The strategy proposed in this study is expected to be applicable to the investigation of the molecular mechanisms of these natural small compounds. Our study examines changes in transcriptome expression patterns, proposing potential targets for the action of GA. Based on these findings, future studies may clarify the regulatory mechanisms of GA, or explore various phytochemicals such as GA using the methodological strategies employed in this study. Our findings may provide valuable clues not only for understanding the action mechanisms of phytochemicals but also for identifying potential side effects.”
To explain how to evaluate the Zhang score, mentioned in the text or as the legend to table 1.
[Response] Thank you for your comments. We have added an explanation for the Zhang score in the main text and as the footer for Table 1 (Page 10, line 349; Page 6, line 162).
“In this method, a query signature (GA treatment) is compared to each of the drug-gene expression profiles ranked in the LINCS L1000 database based on their respective scores. The score ranges between -1 (representing the highest negative correlation) to 1 (representing the highest positive correlation), implying that a higher score indicates greater similarity. Target proteins were restricted to antagonists or inhibitors with pharmacological activity.”
“The Zhang score (connectivity score) was used as a measure of similarity. A higher score indicates greater similarity. Target proteins were restricted to antagonists or inhibitors with pharmacological activity.”
We found in the review of the manuscript that there is an error in the view of the Supplementary Materials, suggesting that you resolve this inconvenience.
[Response] Thank you for pointing this out. It seems like the Supplementary Materials were missing. We apologize for any inconvenience that this may have caused. We have attached the revised Supplementary Materials after careful review and making the necessary corrections.
Regarding the conclusions, it would be indicated to certify in more detail that gallic acid can be a model to support the elucidation of the action mechanisms of other natural compounds with therapeutic potential.
[Response] We appreciate your suggestion. We have now included in-depth details in the main text (Page 8, line 268; Page 10, line 373)
“This study aims to propose a methodology for investigating natural compounds with multifaceted and comprehensive therapeutic effects such as GA. GA, a polyphenol compound found in various plants, is renowned for its potent antioxidant and anti-inflammatory properties, contributing to therapeutic activities in cardiovascular diseases, cancer, neurodegenerative disorders, and aging [1]. Consequently, GA and its derivatives are often used as promising lead compounds for new drug development, contributing to drug modeling and medicinal chemistry research [47,48]. However, research on the mechanisms of GA has been limited due to its complexity. Recently, useful data sources, such as the LINCS database, became available for interpreting mechanisms of action, allowing us to conduct this study. The strategy proposed in this study is expected to be applicable to the investigation of the molecular mechanisms of these natural small compounds. Our study examines changes in transcriptome expression patterns, proposing potential targets for the action of GA. Based on these findings, future studies may clarify the regulatory mechanisms of GA, or explore various phytochemicals such as GA using the methodological strategies employed in this study. Our findings may provide valuable clues not only for understanding the action mechanisms of phytochemicals but also for identifying potential side effects.”
“The availability of valuable data sources, such as the LINCS database, has recently enabled such study, proposing a new strategy that may be applicable to the investigation of the molecular mechanisms of these natural small compounds.”
Round 2
Reviewer 1 Report
Comments and Suggestions for Authors
Enclosed
Reviewer 2 Report
Comments and Suggestions for Authors
Thank you for taking into account my suggestions and answering the 8 questions constructively, thus improving the quality of the manuscript.
Yes indeed the Supplementary Materials can be viewed.
​